# The Biochemical Alteration of Enzymatically Hydrolysed and Spontaneously Fermented Oat Flour and Its Impact on Pathogenic Bacteria

**DOI:** 10.3390/foods11142055

**Published:** 2022-07-12

**Authors:** Paulina Streimikyte, Jurgita Kailiuviene, Edita Mazoniene, Viktorija Puzeryte, Dalia Urbonaviciene, Aiste Balciunaitiene, Theodore Daniel Liapman, Zygimantas Laureckas, Pranas Viskelis, Jonas Viskelis

**Affiliations:** 1Lithuanian Research Centre for Agriculture and Forestry, Institute of Horticulture, 54333 Babtai, Lithuania; paulina.streimikyte@lammc.lt (P.S.); viktorija.puzeryte@lammc.lt (V.P.); dalia.urbonaviciene@lammc.lt (D.U.); aiste.balciunaitiene@lammc.lt (A.B.); pranas.viskelis@lammc.lt (P.V.); 2Roquette Amilina, 35101 Panevėžys, Lithuania; jurgita.kailiuviene@roquette.com (J.K.); edita.maizoniene@roquette.com (E.M.); 3Faculty of Medicine, Riga Stradins University, LV-1007 Riga, Latvia; ted.liapman@rsu.edu.lv; 4Faculty of Medicine, Lithuanian University of Health Sciences, 44307 Kaunas, Lithuania; zygimantas.laureckas@lsmu.stud.lt

**Keywords:** enzymes-assisted extraction, sugars profile, HPLC-SEC, fermentation, antimicrobial properties, *A. sativa*, birch sap, Tibetan kefir grains, INFOGEST in vitro digestibility

## Abstract

*Avena sativa* (*A. sativa*) oats have recently made a comeback as suitable alternative raw materials for dairy substitutes due to their functional properties. Amylolytic and cellulolytic enzyme-assisted modifications of oats produce new products that are more appealing to consumers. However, the biochemical and functional alteration of products and extracts requires careful selection of raw materials, enzyme cocktails, and technological aspects. This study compares the biochemical composition of different *A. sativa* enzyme-assisted water extracts and evaluates their microbial growth using spontaneous fermentation and the antimicrobial properties of the ferment extracts. Fibre content, total phenolic content, and antioxidant activity were evaluated using traditional methodologies. The degradation of *A. sativa* flour was captured using scanning electron microscopy (SEM); moreover, sugar and oligosaccharide alteration were identified using HPLC and HPLC-SEC after INFOGEST in vitro digestion (IVD). Additionally, taste differentiation was performed using an electronic tongue with principal component analysis. The oat liquid extracts were continuously fermented using two ancient fermentation starters, birch sap and Tibetan kefir grains. Both starters contain lactic acid bacteria (LAB), which has major potential for use in bio-preservation. In fermented extracts, antimicrobial properties against Gram-positive *Staphylococcus aureus* and group *A streptococci* as well as Gram-negative opportunistic bacteria such as *Escherichia coli* and *Pseudomonas aeruginosa* were also determined. SEM images confirmed the successful incorporation of enzymes into the oat flour. The results indicate that using enzyme-assisted extraction significantly increased TPC and antioxidant activity in both the extract and residues. Additionally, carbohydrates with a molecular mass (MM) of over 70,000 kDa were reduced to 7000 kDa and lower after the incorporation of amylolytic and cellulolytic enzymes. The MM impacted the variation in microbial fermentation, which demonstrated favourable antimicrobial properties. The results demonstrated promising applications for developing functional products and components using bioprocessing as an innovative tool.

## 1. Introduction

The presence of additives and higher sugar content in food products has increased the development of functional and clean food categories [1]. However, many challenges remain due to the demand for stable products and components, functionality, and overall quality. Oats (*Avena sativa*) are a well-known grain with a wide variety of applications. They are mainly known for its high content and variety of polysaccharides, such as beta-glucans and phenolic compounds, and are a typical breakfast for the nourishment of healthy and obese patients alike. Moreover, oats and their morphological parts are widely processed grains for nondairy substitutes, plant proteins, fermented food, and additives such as sweeteners (e.g., xylitol from oat hull) and thickeners [2,3]. Enzymes for grain material are commonly used to obtain fermentable sugars; specifically, controlled enzymatic hydrolyses release derivates. Polysaccharides cleaved into smaller monomers are attractive due to their novelty and sensory appealing properties [4]. Oat-based drinks (OBD) are gaining popularity and are usually produced via enzymatic hydrolysis due to the resulting reduction in viscosity and increase in sweetness [5,6]. Moreover, α-amylase treatment for oat flour resulted in more DPPH^•^ and FRAP antioxidant activity than heating treatment. The number of phenolics such as avenanthramide 2c, 2p, and 2f, gallic and caffeic acids, and vanillin also increases significantly with this treatment [7].

Today, 69% of consumers believe that plant-based beverages are suitable for their kids, and 77% of millennials in the US drink plant-based drinks regularly [8]. For the most part, health requirements are covered by the consumption of dietary fibres and of less fat and sugar. However, the glycaemic index (GI) of plant-based beverages is medium or higher and varies between 47 and 99 [9]. According to Atkinson et al. [10], GI from 56 to 69 is considered medium; below 55 is low, and above 69 is high, and protein, fat, and fibre content all impact GI. During digestion, various salts impact the isomerization and charges of sugars, leading to altered gut balance and the mucosal absorption of intestinal epithelium [11].

Fermentable sugars produced via enzymatic hydrolysis targeting carbohydrates are economically suitable for many applications. These applications include fermentation using various bacterial strains. Fermentable oligosaccharides, disaccharides, monosaccharides, and polyols, known as FODMAPs, play an essential role in fermentation [12]. In some studies, enzymatically hydrolysed oat hull was fermented using *Spathaspora hagerdaliae* UFMG-CM-Y303, *Saccharomyces cerevisiae* strains, and other yeasts to produce xylitol, a sugar alcohol, and ethanol [3,13,14]. Combined saccharification and simultaneous or spontaneous fermentation are also used to produce fermented food and beverages, especially in beer brewing [11,15]. On top of that, the benefit of lactic acid bacteria (LAB) fermentation for the gut microbiota is a highly contemporaneous subject that is commonly associated with the prevention of obesity and diabetes as well as with better cognitive function [12,16,17]. According to Bocchi et al. [17], oat beverages fermented with lactic acid bacteria strains increased α-tocopherol and d-tocopherol and reduced the number of lignans such as lariciresinol, meairesinol, and secoisolariciresinol [17]. Oat solid-state fermentation with *Lactobacillus palntarum* and *Phizopus oryzae*, according to Wu et al. [18], increased crude protein content by 23–29 mg/g as well as essential and nonessential amino acids, calcium, and potassium [18].

Fermented food has been consumed since 6000 BC, and fermentation is an effective way of preservation [19]. However, metabolites produced by fermenting strains such as organic acids, bacteriocins, and antimicrobial peptides may also have impacts on antimicrobial properties in the human context. This effect is emerging due to the development of rapid antibiotic resistance in the last decade [11,20]. Among the infectious bacteria commonly treated with antibiotics, two are the most abundant in the human population: Gram-positive *Staphylococcus aureus* (*S. aureus*) and Gram-negative *Escherichia Coli* (*E. Coli*) [21]. The widely spread *S. aureus* can cause a broad spectrum of infections including skin and soft-tissue infections as well as foodborne illnesses, which are rapidly developing antibiotic resistance against which new prevention methods are required [22,23]. Similarly, antibiotic resistance for most available antibiotics has developed in *E. Coli*, a pathogen that is often responsible for sepsis and urinary tract infections. The O157:H7 strain of *E. Coli* is a high-risk foodborne pathogen with severe disease presentation [24,25].

Tibetan kefir grains (TKG) and birch sap are ancient known ingredients in the production of fermented products. TKG is a complex symbiosis of microbes and yeasts consisting of LAB, which is used for kefir production [26,27,28]. In contrast, birch sap is a semi-sweet nutritious drink collected from the birch tree [29] that if not thermally processed, ozonized, or iced in a few days following collection will start to naturally ferment [30,31]. Fresh birch sap is a nutritious medium for lactic acid bacteria growth and for the enrichment of sugars or co-substrates as suggested by Semjonovs et al. [32]. Both ingredients carry benefits and contribute to human health. However, there is still a lack of scientific knowledge about spontaneous symbiotic fermentation in plant-based beverages.

This study compared different enzymatic hydrolytic reactions in oat-based drink production by continuously fermenting these extracts using the spontaneous fermentation starters TKG and birch sap. Biochemical, physicochemical, and functional alterations were evaluated as described in the graphical overview in Figure 1. Moreover, OBDs were introduced to a variety of static in vitro digestion-simulating environments to better understand *A. sativa* FODMAPs. Additionally, for fermented drinks, microbial analyses were performed, and antimicrobial properties were evaluated.

## 2. Materials and Methods

### 2.1. Chemicals

The following substances and solvents were used in this study: ethanol 96% (*v*/*v*), (AB Strumbras, Kaunas, Lithuania), Folin–Ciocalteu reagent, gallic acid (3,4,5-trihydroxybenzoic acid, 99%), DPPH^•^ (2,2-diphenyl-1-picrylhydrazyl hydrate free radical), Trolox (6-hydroxy-2,5,7,8-tetramethyl-chroman-2-carboxylic acid), Na_2_CO_3,_ potassium acetate, acetic acid, TPTZ (2,4,6-Tris(2-pyridyl)-s-triazine) (Carl Roth, Karlsruhe, Germany), iron (III) chloride hexahydrate (Vaseline-Fabrik Rhenania, Bonn, Germany), DMCA (4-(dimethylamino)-cinnamaldehyde), neutral detergent solution (ANKOM, Macedon, NY, USA), Sodium sulfite—Na_2_SO_3_, anhydrous (FSS, ANKOM Technology, City, Macedon, USA), and cetyl trimethylammonium bromide (CTAB) (FSS, ANKOM Technology, Macedon, NY, USA).

### 2.2. Plant Material (Dependent Sub-Sample)

The study was performed using an ecological cereal of oat flakes (*A. sativa*) harvested in Lithuania in 2021 (AUGA, Vilnius, Lithuania). Three 1 kg batches were prepared from stock by the supplier for the experiment. Prior to the extraction, plant materials were ground with an ultra-centrifugal mill, ZM 200 (Retsch, Haan, Germany), using a sieve with 0.5 mm holes.

### 2.3. Enzyme Products (Independent Sub-Sample)

GRAINZYME NL is a classical multienzyme product used as an adequate substitute for lowering viscosity in grain products. The cocktail’s main enzyme component is cellulase. The product is reported to have cellulase activity of 5000 U mL^−1^. However, it is also known to contain various additional active substances including hemicellulase, endo-xylanase, exo-xylanase, beta-glucanase, mannanase, galactosidase, and pectinase (Suntaq, Guangzhou, China).

SQzyme GL is a glucoamylase monocomponent [EC3.2.1.3] derived from the fermentation of wild Aspergillus niger (Suntaq, China). The product is reported to have a glucoamylase activity of 150,000 U mL^−1^.

SQzyme BAL is a food-grade bacterial α-amylase [EC3.2.1.1] derived from the fermentation of wild Bacillus subtilis (Suntaq, China). The product is reported to have a glucoamylase activity of 180,000 U mL^−1^.

### 2.4. Static In Vitro Digestion

The static in vitro digestion (IVD) of enzymatically hydrolysed hydrophilic extracts of *A. sativa* was carried out in three different stages as described by Minekus et al. [33]. Modifications to the procedure, as described by Streimikyte et al. in a study conducted in 2020, were incorporated. [34] Briefly, 5 mL of sample with 2 g of glass beads were mixed with simulated saliva fluids (SSF) and pre-incubates at 37 °C for a few minutes. Amylase was added, and the mixture was incubated for 2 min in a rotary mixer. After the samples were mixed with simulated gastric fluid (SGF) containing pepsin (2000 U mL^−1^ of digest), the pH was adjusted to 3 using 5 mol L^−1^ of HCl. The final digest volume was adjusted to 20 mL. The mixtures were placed into a temperature-controlled thermostat with a continuous rotator at 600 rad s^−1^. After two hours of incubation, the final intestinal step was carried out by adding simulated intestinal fluid (SIF) supplemented with the following individual enzymes: trypsin (100 U mL^−1^ of di-gest), chymotrypsin (25 U mL^−1^ of digest), pancreatic amylase (200 U mL^−1^ of digest), porcine pancreatic lipase (2000 U mL^−1^ of digest), and bile salts (10 mM of di-gest). The pH was adjusted to 7 by adding 5 mol L^−1^ of NaOH. The final volume of the sample was 40 mL. IVD was then performed for 180 min. The digestion process of the samples was stopped at gastric phase points by adjusting the pH to 7 and by cooling the samples; the digestion process for the intestinal phase samples was stopped only by cooling the samples to 0–4 °C in ice water. After cooling, the samples were centrifuged at 10,000 rpm at +4 °C and filtered. The soluble fraction of the digest was collected, frozen, lyophilized, and stored at +6 °C prior to analysis. The digestion procedure was performed twice.

### 2.5. Enzyme-Assisted Extraction and Spontaneous Fermentation Using Tibetan Kefir Grains and Birch Sap

Enzyme-assisted extraction was performed with slight modifications as described by Streimikyte et al. [35]. Two different batches were analysed. Milled oat cereals (>0.5 mm) were homogenized with distilled water at a ratio of 1:5. Following that, 0.15% of AL and 0.15% NSP were added to the first batch, and 0.45% AL + AG, 0.15% NSP was added to the second batch. After 2.5 h of enzymatic extraction at 68 °C, liquid and solid fractions of the oats were collected. The enzymes were inactivated using heat treatment at 95 °C for 5 min. The liquid and solid parts were separated using a 100-micron sieve and then frozen and freeze-dried respectively until analysis. Following the enzyme-assisted extraction, but before the freezing, the liquid phase was used for further fermentation. Specifically, 10% of Tibetan Kefir grains were incorporated into the liquid samples. In parallel, the oat liquid fraction was homogenized with fresh birch sap at a ratio of 1:1. Fermentation was carried out in an incubator at a fixed temperature of 28 °C for five days. After five days, the samples were filtered and frozen at −20 °C until analysis.

### 2.6. Fibre Analysis

The analysis of Neutral Detergent Fibre (NDF) and acid detergent fibre (ADF) was performed. An ANKOM^2000^ fibre analyser (ANKOM Technology, Macedony, NY, USA) was used to evaluate cellulose, hemicellulose, and lignin content. Both methodologies are described in detail at https://www.ankom.com/analytical-methods-support/fibre-analyser-a200 (last visited 21 May 2022). ADF represents the remaining fibre content following digestion with H_2_SO_4_ and CTAB; it estimates the final content of cellulose, hemicellulose, and lignin. For the analysis, *A. sativa* lyophilized samples were prepared. Furthermore, 0.45–0.50 g of *A. sativa* flour samples were sealed in filter bags. The samples were placed in a bag suspender and inserted into the fibre analyser for processing. The instrument then performed all the necessary steps for the digestion and rinsing of the samples. The samples were subsequently dried and weighed. The ADF content was calculated using the following formula:(1)$$\mathrm{ADF}\left(\%\right)=\frac{100\times \left(W_{3}-\left(W_{1\;}\times C_{1}\right)\right)}{W_{2}}$$

W_1_—bag tare weight; W_2_—sample weight; W_3_—dried weight of the bag containing fibre following the extraction process, C_1_—blank bag correction (running average of the final oven-dried weight divided by original blank bag weight).

The neutral detergent fibre method was used to evaluate the remaining residue after digestion in a detergent solution. The fibre residue predominantly comprised hemicellulose, cellulose, and lignin. The samples were prepared using the same steps as ADF. For the extraction, an additional 4 mL of alpha-amylase with an activity level of 17,400 Liquefon units mL^−1^ (FAA, ANKOM Technology) as well as 20 g of sodium sulfite were added. After the extraction, the filter bags containing the samples were washed with acetone and oven-dried. NDF content was calculated using the following formula:(2)$$\mathrm{NDF}\left(\%\right)=\frac{100\times \left(W_{3}-\left(W_{1\;}\times C_{1}\right)\right)}{W_{2}}$$

W_1_—bag tare weight; W_2_—sample weight; W_3_—dried weight of the bag containing fibre following the extraction process, C_1_—blank bag correction (running average of the final oven-dried weight divided by original blank bag weight).

### 2.7. Scanning Electron Microscopy (SEM) Analysis

The morphological structure of the tested plant was examined using the images obtained with SEM FEI Quanta 200 FEG (FEI Company, Hillsboro, OR, USA). The samples were analysed in low vacuum mode operating at 3.0 kV using an LDF detector. *A. sativa* powder samples from the control and enzymatically hydrolysed residue groups were spread on an aluminium table and measured at three different locations. 

### 2.8. Determination of Total Phenolic Content

Total phenolic content was determined with the Folin–Ciocalteu method with slight modifications as described by Viskelis et al. [36] using gallic acid as the standard [37]. The absorbance of the samples was measured at 765 nm using a Cintra 202 (GBC Scientific Equipment, Knox, Braeside, Australia) spectrophotometer. The total phenolic content was calculated from a gallic calibration curve and was expressed as mg per 100 g gallic acid equivalent (GAE) per gram of dry weight (mg GAE 100 g^−1^).

### 2.9. Determination of Antioxidant Capacity

DPPH^•^ free radical scavenging activity was determined using the method proposed by Brand Williams, Cuvelie, and Berset [38] with slight modification [39]. Specifically, 2 mL of a DPPH (2,2-diphenyl-1-picrylhydrazyl) solution in 99.0% *v*/*v* ethanol was mixed with 20 μL of *A. sativa* extract samples. A decrease in absorbance was recorded at 515 nm with a Cintra 202 (GBC Scientific Equipment, Knox, Braeside, Australia) spectrophotometer after 30 min.

The ferric-reducing antioxidant power (FRAP) assay was carried out as described by Benzie and Strain [40] with some modifications. The FRAP solution consisted of TPTZ (0.01 M dissolved in 0.04 M HCl), FeCl_3_ × 6H_2_O (0.02 M in water) and acetate buffer (0.3 M, pH 3.6) at the ratio of 1:1:10. Then, 2 mL of freshly prepared FRAP reagent was mixed with 2 μL of *A. sativa* extract samples. An increase in absorbance was recorded at 593 nm with a Cintra 202 (GBC Scientific Equipment, Knox, Australia) spectrophotometer after 30 min.

All antioxidant activity assays were calculated using Trolox calibration curves and expressed as μmol of the Trolox equivalent (TE) per one gram of dry weight (µmol TE g^−1^).

### 2.10. Sugar and Oligosacharide Molecular Mass Profiles

#### 2.10.1. Sugar Identification Using High Pressure Liquid Chromatography (HPLC)

Sugar identification and quantification were performed using an HPLC Dionex Ultimate 3000-4 (International Equipment Trading Ltd., Mundelein, IL, USA) equipped with a column oven and an integrated Aminex HPX-87H column (300 × 7.8 mm) (Bio-Rad, Hercules, CA, USA). After separating the analytes from the mixture on the analytical column, the individual compounds were then detected using UV and IR detectors (Wyatt, Santa Barbara, CA, USA). The following main parameters were applied: 5 M H_2_SO_4_ eluent, a flow rate of 0.4 mL min^−1^, temperature of 60 °C, work pressure of 38–40 bar, and injection volume of 20 μL. The Sugar in the samples was hydrolysed using 60% H_2_SO_4_

#### 2.10.2. High Pressure Liquid Chromatography Size Exclusion for Molecular Mass Distribution (HPLC-SEC)

A Dionex Ultimate 3000 HPLC (International Equipment Trading Ltd., Mundelein, IL, USA) equipped with a column oven with integrated size-exclusion Ohpak SB-802 HQ (8 × 300 mm (8 μm)) (Shodex, Munich, Germany) and Ultrahydrogel 500 (7.8 × 300 mm (10 μm)) (Waters, Wilmslow, UK) columns. Sugars were detected with the RI detector Optilab T-rEX (Wyatt, Santa Barbara, CA, USA). As a standard, in order to identify the degree of polymerisation (DP), the following monomeric units were used: DP1-D-(+)-glucose (≥99.5%; G8270; Sigma, St. Louis, Missouri, USA); DP2-D-(+)-mannose (≥95%; 92683; Supelco, Bellefonte, PA, USA); DP3-maltotriose (>99%; GLU313; Elicityl S.A, Crolles, France) DP4-maltotetraose (>99%; GLU314; Elicityl S.A, Crolles, France). The following main parameters were applied: 0,05 M NaNO_3_ eluent, a flow rate of 0.5 mL min^−1^, temperature of 40 °C, work pressure of 50–52 bar, RI detector, and an injection volume of 50 μL. A sample solution containing 2 to 20 mg g^−1^ (1% DMSO-dimethylsulphoxide as internal standard) of lyophilized matter was used for the test. The samples were stirred with a magnetic stirrer for 30 min. The required amount of sample (~1 mL) was filtered through a 0.45 μM syringe filter into a chromatographic flask (the filtrate had to be precise). Results were obtained by analysing the chromatograms. In this case, the retention times (tR) of each peak and internal standard (DMSO) were essential. These values and the chromatogram data were exported in the csv format and other required data (baseline and coordinates) were generated in a standard calculation file. The molecular weight distribution curve was plotted against sample concentration curves based on regularly checked and updated calibrations.

### 2.11. Electronic Tongue for A. sativa Extracts and Commercial Oat Drinks

A potentiometric electronic tongue (ET) produced by αAstree (Alpha M.O.S., Toulouse, France) was used. The device consisted of four parts, including a 48-tray liquid automatic sampler system, 7-liquid cross-selective taste sensors (ZZ, JE, BB, CA, GA, HA, and JB), an Ag/AgCl reference electrode, and a PC computer with ASTREE Alpha M.O.S V12 software to perform statistical analyses. The sensor array is composed of 7 solid potential sensors that are a chemically modified field of effect transistors (ChemFET). These sensors are coated with a specific membrane (chemical compounds) to induce both cross-sensitivity and cross-selectivity. Product samples were measured with ET according to a procedure protocol obtained from Alpha MOS. The samples were centrifuged at 9000 × g for approximately 10 min. Nine measurements of each sample were performed for the data analysis, which was carried out using PCA to determine and characterize the correlations among the tastes of samples 2—batch I: *A.sativa* hydrophilic extract; 3—batch I: *A.sativa* hydrophilic extract diluted in water 80:20; 4—batch I: *A.sativa* hydrophilic extract diluted in water 70:30; 5—batch II: *A.sativa* hydrophilic extract; 6—Commercial OBD_1; 7—Commercial OBD_2; 8—Commercial OBD_3; 9—Commercial OBD_4; 10—Commercial OBD_5. The characteristics of the used commercial products that were produced using the different treatments are listed under Appendix A. PCA (Figure 6). The PCA method is an unsupervised classification method that has been used successfully in many applications [41].

### 2.12. Microbial Evaluation of Fermented Samples

The microbial evaluation of the spontaneously fermented samples with TKG and birch sap was performed using agar diffusion to determine the growth of *Lactobacillus delbrueckii* subsp. *bulgaricus*, and *Streptococcus thermophilus*. Wells of 6 mm diameter were punched in the agar and filled with the fermented extracts. Agar plates were incubated at 37 °C for 24 h, and the colonies were counted and expressed in cfu/mL. Viable mesophilic lactic bacterial counts were established by means of serial dilution and subsequent plating.

### 2.13. Antibacterial Activity Assay

Antibacterial activity in vitro was evaluated using agar diffusion against Gram-positive group A beta-hemolytic *streptococci*, *Staphylococcus aureus*, and Gram-negative *Pseudomonas aeruginosa* and *Escherichia coli* bacteria strains. For this purpose, sterile cotton was used. A suspension (~10^8^ cfu/mL) of bacterial strains at 0.5 McFarland unit density was inoculated onto the cooled Mueller Hinton Agar (Oxoid, Basingstoke, UK) using swabs. Wells of 6 mm in diameter were punched in the agar and filled with 50 µL of extracts. The agar plates were incubated at 37 °C for 24 h, and zones of inhibition were measured and tabulated. 

### 2.14. Statistical Analysis

All the results were presented as means ± standard deviation (SD), and all experiments were performed at least three times. In addition, one-way ANOVA followed by Tukey’s HSD test were calculated to compare the means and demonstrated significant variation (*p* < 0.05). This was performed and calculated with the statistical package GraphPad Prism 8 software (GraphPad, San Diego, CA, USA).

## 3. Results and Discussion

### 3.1. Hemicellulose, Cellulose, and Lignin Content

Fibre content was analysed before and after the enzymatic hydrolysis of *A. sativa*. Table 1 shows the hemicellulose, cellulose, and lignin content after acid detergent fibre (ADF) analysis and neutral detergent fibre (NDF) analysis. The results indicated that hemicellulose, cellulose, and lignin content in the hydrolysed fractions were 15.41–12.27% lower than in the control samples. This indicates that the non-starch polysaccharide enzymes (NSP) have hydrolysing properties. As the results show, 22.40% of *A. sativa* control samples consisted of hemicellulose. The *A. sativa* contents in batches I and II were 12.62% and 6.67%, respectively. This indicates that glucoamylase (GL), which was incorporated in batch II, cleaves the released polysaccharides and oligosaccharides into smaller monomers [15]. Additionally, because the *A. sativa* flour was stripped of the husk and bran and milled into fractions less than 0.5 mm in size, the enzymes were able to achieve higher activity. In comparison, for wheat straw, the release of hemicellulose reached 96 g/kg to 135 g/kg using enzyme cocktails with higher xylanolytic enzyme concentrations and 42.5–59.8% extraction [42].

### 3.2. Scanning Electron Microscopy (SEM)

Images of enzymatically extracted spent grain were taken using SEM. As shown in Figure 2, the control sample of oat flour shows compact, well-shaped structures of milled flour. In comparison, in the enzyme-treated samples, there was an increase in spherical-shaped structures, which were clearly observed in samples (c) and (d). These findings correlate with those of Zhang et al. [43], who identified oat starch groups that seemed irregular and dispersed using SEM. The structures in Figure 2 demonstrate that enzymes can access and disrupt polymeric chains into smaller ones. 

The main non-starch polysaccharide enzyme is cellulase (EC 3.2.1.4). However, the manufacturer (see “Materials and Methods”) states that potential activity of endo- and exo-cellulase, beta-glucanase, xylanase, and others may be present. This may result in synergistic activity with amylolytic enzymes such as amylase and glucoamylase, leading to rapid cleaving of the glycosidic linkages of the complex matrix of oat flour. The complex matrix can be seen in Figure 2a. For example, amorphous cellulose regions possess a higher affinity for cellulase because of the synergy with xylanases, as described in wheat straw enzymatic hydrolysis [42]. Moreover, NSP and starch-degrading enzyme mixture incorporation is common in brewing; NSP enzymes degrade the endosperm cell wall to increase the viability of starch, which is degradable by amylases and improves mashing [15]. In contrast, SEM pictures of the ultrasound-assisted extraction of oat bran at 45 Hz showed increased permeability but a consistent structure that was not seen in enzyme-assisted extraction [44]. 

### 3.3. Total Phenolic Content (TPC) and Antioxidant Activity through Enzymatic Extraction Kinetics

The enzymatic extraction kinetics were analysed for total phenolic content, DPPH•, and FRAP in in vitro antioxidant activity. The kinetics shown in Figure 3 represent similar asymptotic behaviour, reaching a plateau at around 150 min [42,45]. Enzyme assistance is widely applicable for enhancing total phenolic content, antioxidant activity, and peptide number [46,47,48,49]. A. sativa enzyme usage is recommended due to the residual starch particles, which may lower the extraction of phenols [44]. The samples in Figure 2 and Table 1 were extracted using α-amylase (AL), GL, and NSP cocktails. 

TPC content increased 2.3 times from the beginning of the time enzyme-assisted hydrolysis, and FRAP antioxidant activity increased from 1.1 to 4.9 µM TE 100 mL^−1^. Following that, spent grain was collected after the enzyme-assisted extraction of oat flour, lyophilized, and re-extracted using different extraction solvents. The TPC content was measured, and the results are shown in Table 2. The results indicate that the highest extraction yield was achieved in enzymatically hydrolysed residues with acetone solvent 30:30 (*v*/*w*). In general, most samples reduced the content of phenols. However, it may explain the extractable content in the hydrophilic phase during enzyme-assisted extraction.

Studies show that a temperature of 70 °C is preferable as it doubles the phenolic content from *A. sativa*. This is similar to our findings for enzymatic extraction. Moreover, UAE can be implemented to enhance the TPC content in oat bran. In a study conducted by Chen et al., [50] TPC composition using UAE increased 1.5-fold over conventional extraction methods. Although avenanthramides are involved in many metabolic pathways like COX2/PGE2 pathways and induce NK-κB inactivation in C2C12 cells, there are many other phenolic compounds that include gallic acid, anthranilic acid, and syringic acid and that have health-promoting properties. For example, syringic acid has anticancer properties in vivo and in vitro by potently inhibiting the proliferation of SW-480 colorectal cancer cells [51,52].

Oat pomace after extraction can be valuable material for further extractions of phenolic content with high antioxidative properties and inhibitory effects on OA-induced fatty liver model in vitro [53]. Polyphenol avenanthramides, explicitly found in oats, have been proven to protect from oxidative stress and regulate the nuclear factor erythroid 2-related factor (Nrf2) signalling pathway in P12 cells, which is a promising neuroprotective agent [54]. Re-extracted phytochemicals and disturbed long-chain polysaccharides were hydrolysed, resulting in a mixture of fermentable sugars and phytochemicals. These can also be incorporated into the medium as reducing or capping agents used in the green synthesis of nanoparticles [55,56]. Oat residues, after extraction, can also be referred to as brewer’s spent grain (BSG) due to the similar saccharification processes applied in OBD production. BSG is the main solid waste after principal extraction. Recent studies suggest that spent grain use is appropriate for enzyme-cocktail production using solid-state fermentation, reducing expenses for the substrate by producing enzymes that lower the cost by up to 30% [57,58]. 

### 3.4. Sugar Profile and Variation after Static In Vitro Digestion

#### 3.4.1. Sugar Identification Using High-Pressure Liquid Chromatography (HPLC)

The quantitative profiles of the monomeric sugars and organic acids are represented in Figure 4. Glucose is the main simple sugar in these samples, making up to 94.5 ± 0.9%, while the composition of other sugars varied between 0.2% and 2.0% in enzyme-assisted extracts. Similar tendencies were observed in samples after simulated in vitro digestion, where glucose monomeric units dominated at 92.5 ± 1.45% compared with other saccharides, which varied between 0.3% and 5.1%. Compared with the control extracts prepared without enzymes, there were more significant increases of 1.19 ± 0.04 and 1.43 ± 0.09 in the sugar content after hydrolysis for the first and second batches, respectively. However, as shown in Figure 4, digestive fluids and electrolytes impacted the compositions of the different sugars. Total sugar quantity was significantly higher after the IVD of batch I extract. Moreover, trisaccharides (named “DP3” in Figure 2) in samples without glucoamylase have not been affected by digestive enzymes, in contrast with batch II. This suggests that digestive enzymes specifically target regions of carbohydrates that were already disturbed by bacterial enzymes. For the most part, sugar profiles and fermentability are essential to brewing beer [3,59].

Studies have shown that wheat mort (maltose, the main simple sugar) levels reach 45%, DP3 levels reach 15%, and glucose levels reach 10%. The same studies show that barley wort contains 54–60 g L^−1^ of maltose [15,59]. However, the analysis in this study demonstrates that most of the maltose was hydrolysed to glucose monomeric units with 60% H_2_SO_4_. The remaining maltose accounted for <3% of all the identified simple sugars in *A. Sativa* enzyme-assisted hydrolysates.

#### 3.4.2. High-Pressure Liquid Chromatography Size Exclusion for Molecular Mass Distribution (HPLC-SEC)

HPLC-SEC was performed to evaluate the length of saccharides and see the alteration after static in vitro digestion (IVD). The approach was selected to compare the molecular masses of the sugars and the degree of polymerization (DP). The length of oligosaccharides is known to be 3–9 monomeric units (DP3-DP9). Furthermore, current studies indicate that oat carbohydrates lead to a higher short fatty acid content, which contributes to gut health and has a prebiotic potential [2,60]. As shown in Figure 5, oat control, a liquid phase sample without enzymes that was lyophilized before analysis consists of carbohydrates with a molecular mass of over 70,000 kDa. However, after the incorporation of enzymes, the molecular mass variation decreased to 7000 kDa and below. Studies report that a crystalline structure plays a vital role in starch digestibility [43,61]. GL cleaves smaller oligosaccharides and disaccharides into monomeric units, a crystalline structure, and a double-helix structure, which leads to a higher hydrolysis rate and may cause a higher GI [43]. However, after IVD, the molecular mass of all the samples was reduced to 700 kDa.

No increase in monomeric sugars was seen in Batch I of the hydrolysed oat extract (AS EAE I). This may be explained by the specificity of α-amylase. Another explanation may be rooted in the origin of the used α-amylase that was produced by *T. ressei* strains. For IVD simulation, animal origin α-amylase was used. Similarly, a monosaccharide, a disaccharide, and an oligosaccharide profile were produced by Bocchi et al. [17] for a commercial oat beverage that was digested using an in vitro protocol. In that study, the chromatograph identified higher DP1-DP2 rates that decreased after fermentation and a greater alteration of DP3-DP9 oligosaccharides.

These results demonstrate that digestive enzymes possess excellent access to oat polysaccharides and oligosaccharides. Moreover, AL and NSP, which include xylanases and beta-glucanases, produce higher levels of DP4 and DP3 [62]. These compounds may reduce the hydrolysis rate of IVD and water-soluble beta-glucan, as well as non-starch constituents that reduce the hydrolysis rate [63]. A complex matrix of food products that include proteins and lipids can reduce digestibility [64]. However, more comprehensive studies should be performed. According to the results, the lyophilized oat hydrophilic extract with AL and NSP enzymes is a promising ingredient for enriching food products with oligosaccharides [65].

### 3.5. Electronic Tongue PCA Evaluation of A. sativa Hydrophilic Extracts and Commercial OBD

Achieving the desired organoleptic properties of plant-based beverages depends on the technology and ingredients used, and taste and flavour may differ [66]. Moderate sweetness is one of the key parameters that consumers highly desire [67]. Oat-based beverages are popular and attractive because of their sweetness and soft consistency, which are attributed to their viscosity, which is turn is attributed to beta glucan molecules and released sugars during the production process. This creates similarity to dairy fat [68].

Previous studies have analysed the tastes of different plant-based beverages using electronic tongue devices. However, to the best of the authors’ knowledge, OBDs have never before been analysed using this method. The samples were tested using a commercial electronic tongue (αAstree, Alpha M.O.S.) with the aim of determining taste diversity. PCA (Figure 6) was mapped using the relative voltage responses of the ET sensors to the drink samples. The PCA statistical analysis showed that when using the two first principal components (PC1 and PC2), which covered 69.516% and 27.278% of the variance, the three samples (8, 9 and 10) could be differentiated. No differences were identified between samples 2, 3, 4, 5, 6, and 7. It can be surmised that the ingredients and the technology used, especially enzymatic hydrolysis, are among the main differentiation factors, along with the raw material type. Samples 8, 9, and 10 were separated as during OBD production, and additional additives were incorporated under the premise that they might produce more suitable technological products. However, the incorporation of these additional substances is intended to enhance the flavour and increase the desirability of the product.

**Figure 6 foods-11-02055-f006:**
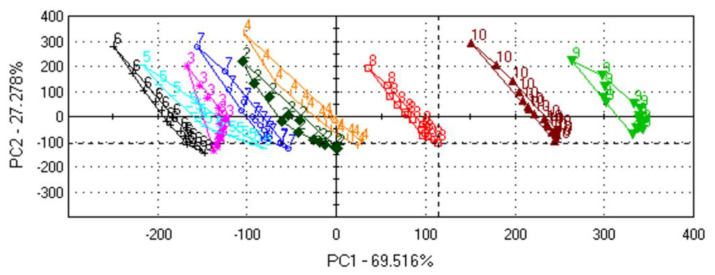
Principal component analysis (PCA) of the samples obtained from the sensory analysis, where 2—batch I: *A. sativa* hydrophilic extract; 3—batch I: *A. sativa* hydrophilic extract 20:80 (*e*/*w*); 4—batch I: *A. sativa* hydrophilic extract 70:30 (*e*/*w*); 5—batch II: *A. sativa* hydrophilic extract; 6—Commercial OBD_1; 7—Commercial OBD_2; 8—Commercial OBD_3; 5—Commercial OBD_4; 9—Commercial OBD_5; 10—Commercial OBD_6 (See the Appendix A).

Nine samples of oat drink beverages were tested, and similarities were identified. The sweetness of OBD is one of the key parameters due to the release of sugar molecules by the enzymatic hydrolysis. The authors suggest that the differences in the PCA plots are due to differences in sweetness. The hydrophilic extracts from batches I and II (named 2, 3, 4, and 5 in the graph) were similar to the commercial products tested and can be considered potential oat-based beverages with appealing flavour.

### 3.6. Microbial Count of Fermented Samples with Tibetan Kefir Grains and Birch Sap

After the enzyme-assisted extraction, hydrophilic extracts were collected and prepared for further fermentation with TKG and BS. After five days of fermentation, samples were filtered, and the quantities of mesophilic lactic acid bacteria, *Lactobacillus delbrueckii* subsp. (*L. delbrueckii*), and *Streptoccocus thermophilus* (*S. thermophilus*) were counted. These microbial strains represent a fraction of the LAB content. *L. delbrueckii* and *S. thermophilus* cultures are used in dairy yogurt production as well as in the fermentation of plant-based yogurts (e.g., with coconut cream, cashew, soy, and almond dairy substitutes) [69]. *S. thermophilus*, in a recent study, demonstrated increased lactic acid and total polyphenol content and DPPH^•^ antioxidant activity in fermented jujube puree [70]. *L. delbrueckii* and *S. thermophilus* are also used for nondairy yogurt production using oat protein concentrate [71]. Table 3 presents classical yogurt bacterial growth using spontaneous fermentation with TKF and BS on differently hydrolysed hydrophilic extracts of *A. sativa* flour. All samples contained more of the mesophilic lactic bacteria *L. delbrueckii* and *S. thermophilus* than 8 log^10^ cfu/mL, indicating that both starters contained LAB. Further on, *A. sativa* samples that were additionally hydrolysed with glucoamylase and were fermented with birch sap (AS II (BS)) contained significantly more *L. delbruckii* and *S. thermophilus* compared with the first batch of hydrophilic extract of *A. sativa* with birch sap. This means that a higher concentration of reducing monomeric sugars increases the LAB content. Similar tendencies were observed in samples fermented with TKG. However, the *L. delbrueckii* count did not show a significant difference between samples AS I (TKG) and AS II (TKG). 

In a recent study, *L. delbrueckii* in soy drink demonstrated stress protein production and inhibited growth [72]. This indicates that the protein origin is essential and that *L. delbrueckii* with *S. thermophilus* degrades oat proteins during fermentation [71]. However, fermentation did not appear to affect the observed textural properties. Nonetheless, the results revealed a promising application of OBD fermentation with TKG and BS. Plant-based beverage fermentation can decrease undesirable compounds like phytates and increase wanted flavour-forming compounds and LAB. In another study, fermented quinoa plant-based beverage reduced phytates by 60.2% after 6 h, increasing zinc and iron absorption [73].

### 3.7. Antimicrobial Properties against Gram—Positive and Gram—Negative Pathogenic Bacterias

It was determined that *A. sativa* extracts have higher phenolic content, antioxidant activity, levels of released sugars, and saccharide lengths, which impacted the growth of selected microbial strains. Extracts were fermented with TKG and birch sap and incubated for five days at a stable temperature of 28 °C. In this study, the antibacterial activity of fermented oat extracts was investigated against both Gram-positive (*A. streptococci* and *S. aureus*) and Gram-negative (*P. aeruginosa* and *E. coli)* bacteria strains (see Table 4). Interestingly, fermented samples with birch sap did not show antibacterial properties against selected strains. However, as shown in Table 4, fermented samples with TKG demonstrated antimicrobial properties against all four selected pathogenic strains. All *A. sativa* samples showed 1.2–5 times more antibacterial activity against Gram-positive bacterial strains than Gram-negative strains. *Lactobacillus kefiranofaciens* comprises over 80% of the microorganisms found in TKG, making it the most abundant. However, more than 700 species have already been identified, 33 of which were probiotic species [74]. LAB is well known for inhibiting the growth of foodborne microorganisms, meaning that plant-based food fermentation is applicable for food biopreservation [75,76].

When comparing antibacterial properties between extracts I and II, the second batch was fermented with more released sugars than the first. This correlates with higher inhibition zones of Gram-positive strains but no significant difference for Gram-negative bacteria. Moreover, during fermentation, bacteriocins, antimicrobial peptides, and other fermentation end-products may impact antibacterial properties [11]. 

The results revealed that *A. sativa* fermented samples with TKG effectively suppress bacterial growth with an inhibition zone between 0.5 and 6.5 mm. Both extract samples showed slightly higher antibacterial activity against Gram-positive bacteria than Gram-negative bacteria (*p* < 0.05) due to the variation in their cell wall structure. As controls, two different antibiotics were used against pathogenic bacteria: for *S. aureus*, penicillin, inhibition zone 26 ± 0.50 mm; for *E.coli*, gentamicin, inhibition zone of 17 ± 0.45 mm.

## 4. Conclusions

The results indicate that there is a significant role to enzyme incorporation in extraction methods. Fibre and starch were disturbed, leading to the release of various lengths of saccharides and organic acids in aqueous extracts with cellulolytic and amylolytic enzymes. Extraction in mild conditions and safe solvents was evaluated as well, and the taste of the hydrophilic extracts of *A. sativa* was assessed using an e-tongue device. This device shows the potential sensory taste perception of beverages. The differentiation of monomeric sugar units and oligosaccharides, as performed in this study, opens the door to possible applications of fermented foods and the production of bio-preservatives.

Additional studies on complex food matrices could lead to future technologies that would allow for stable oligosaccharides after stimulated digestion. Significant applications can be further developed with oat spent grain, which may be appropriate for enzyme production and the purification of functional components. In summary, grain modification using enzyme-assisted extraction creates novel opportunities for developing higher value-added products and comprehensive applications for green synthesis development.

## Figures and Tables

**Figure 1 foods-11-02055-f001:**
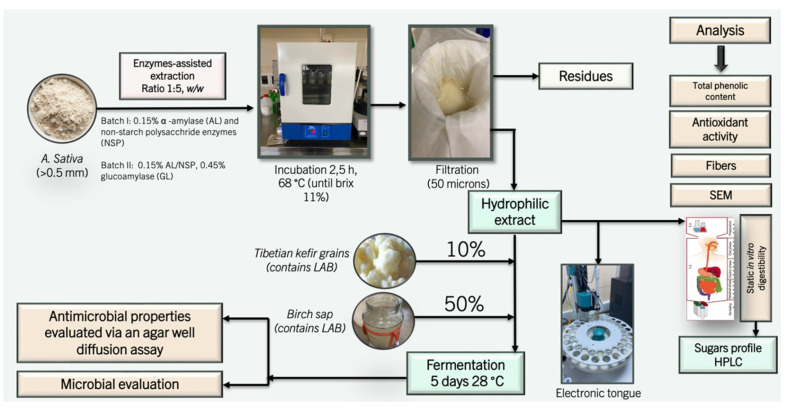
Overview of the experimental design.

**Figure 2 foods-11-02055-f002:**
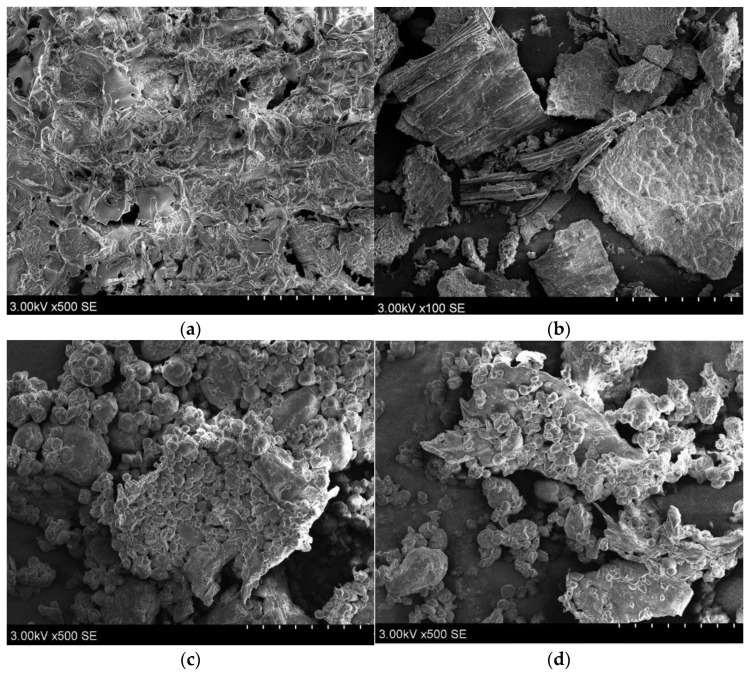
SEM images of *A. sativa* flour before enzyme-assisted extraction (**a**,**b**) and after enzyme-assisted extraction (**c**,**d**). The images are at a scale bar of 100 and 500 µm.

**Figure 3 foods-11-02055-f003:**
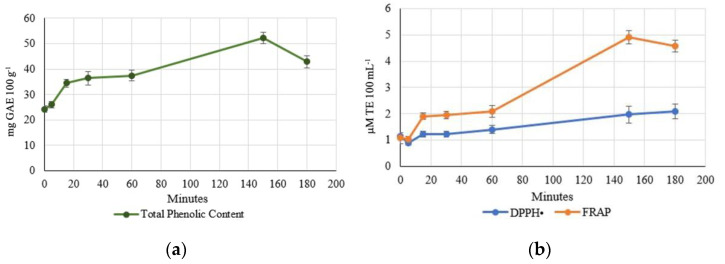
Total phenolic content (**a**) and antioxidant activity and (**b**) the kinetics during the enzyme-assisted extraction of flour during enzyme-assisted water extraction. Values are expressed as mean and standard deviation and were calculated with triplicate determinations.

**Figure 4 foods-11-02055-f004:**
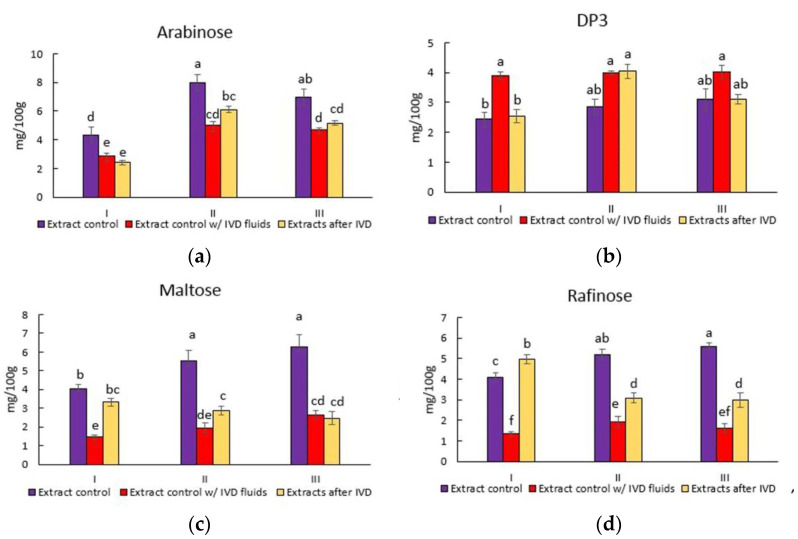
Alterations in monosaccharides, trisaccharides (DP3), total sugars, and total sugars with organic acids (**a**–**i**), where three groups of samples are named I, II, and III. Oat extracts made without enzymes—control (I), oat extracts with an AL and NSP enzyme mixture—batch I (II), and oat ex-tracts with an AL, GL, and NSP cocktail—batch II (III). Values are expressed as mean and stand-ard deviation calculated with triplicate determination; different superscript letters within the same column indicate significant differences (one-way ANOVA and Tukey’s HSD test, *p* < 0.05).

**Figure 5 foods-11-02055-f005:**
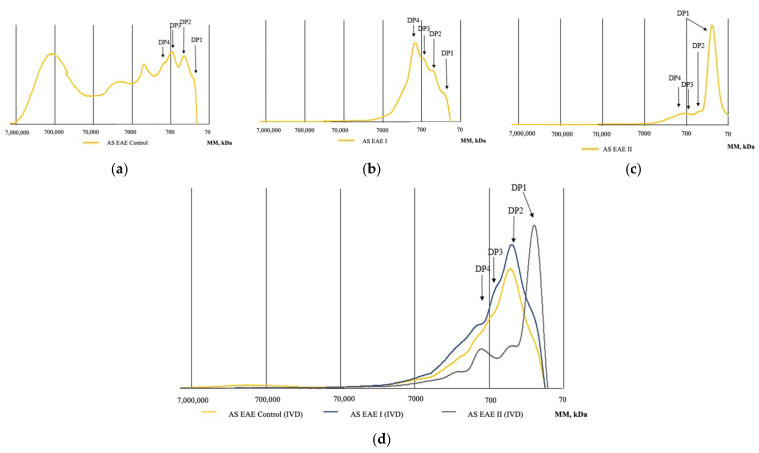
Oat flour molecular mass (kDa) profile produced using HPLC-SEC. Graph (**a**)—oat extract as control (AE EAE Control); Graph (**b**)—batch I oat flour enzyme-assisted extraction (AS EAE I); Graph (**c**)—batch II oat flour enzyme-assisted extraction (AS EAE II); Graph (**d**)—overlay graph of oat flour control, batch I, and batch II after static in vitro digestion. DP1—DP4 represent the degree of polymerization, showing the lengths of the sugars and oligosaccharides.

**Table 1 foods-11-02055-t001:** Fibre content of the *A. Sativa* control and after enzymatic extraction.

Sample Name	Hemicellulose, Cellulose, and Lignin, %	Cellulose and Lignin, %
*A. sativa* control	40.76 ± 1.97 a	18.36 ± 0.51 b
*A. sativa* batch I	28.49 ± 0.85 c	15.87 ± 1.83 b
*A. sativa* batch II	25.38 ± 2.45 c	18.91 ± 3.45 b

Hemicellulose, cellulose, and lignin content after acid detergent fibre (ADF) analysis, and cellulose and lignin content after neutral detergent fibre (NDF) analysis. Batch I: oat extract with an AL and NSP cocktails; Batch II: oat extract with an AL, GL and NSP cocktails. Values expressed as mean and standard deviation of triplicate determinations; different superscript letters within the same column indicate significant differences (one-way ANOVA and Tukey’s HSD test, *p* < 0.05).

**Table 2 foods-11-02055-t002:** The antioxidant activity and total phenolic content of flour before and after enzyme-assisted water extraction using different solvents.

Extraction Solvent and Concentration, % (*s*/*w*)	Before EAE	After EAE
	Oat Flour	Spent Grain
	TPC mg GAE/100 g	
Ethanol		
30:70	66.3 ± 0.21 a	14.5 ± 0.25 b
50:50	80.5 ± 0.15 a	43.6 ± 0.15 b
70:30	70.7 ± 0.26 a	43.7 ± 1.25 b
Methanol		
30:30	60.6 ± 0.52 a	25 ± 0.56 b
50:50	61.9 ± 0.43 a	62 ± 0.26 a
70:30	66.5 ± 1.23 a	51.3 ± 0.88 b
Acetone		
30:30	77.6 ± 1.11 b	91.7 ± 0.22 a
50:50	83.5 ± 0.29 a	71 ± 0.36 b
70:30	82.4 ± 0.89 a	61.7 ± 0.11 b

Extraction solvent and water ratio (*s*/*w*); EAE—enzyme-assisted extraction; TPC—total phenolic content; GAE—gallic acid equivalent. Values are expressed as mean and standard deviation for triplicate determinations; different superscript letters within the same column indicate significant differences (one-way ANOVA and Tukey’s HSD test, *p* < 0.05).

**Table 3 foods-11-02055-t003:** Fermented enzyme-assisted hydrolysate microbiological analysis.

	Mesophilic Lactic Acid Bacteria Count	*Lactobacillus delbrueckii* subsp. *bulgaricus*	*Streptococcus thermophilus*
	cfu/ mL		
*AS* I (BS)	2.3 × 10^8^ d	1.6 × 10^8^ c	1.8 × 10^8^ d
*AS* II (BS)	7.6 × 10^8^ a	5.6 × 10^8^ a	6.7 × 10^8^ a
*AS* I (TKG)	4.7 × 10^8^ c	4.0 × 10^8^ b	4.9 × 10^8^ c
*AS* II (TKG)	5.5 × 10^8^ b	3.9 × 10^8^ b	6.2 × 10^8^ b

Different letters within the same column indicate significant differences (one-way ANOVA and Tukey’s HSD test, *p* < 0.05). AS I (BS)—*Avena sativa* batch I liquid fraction fermented with birch sap; AS II (BS)—*Avena sativa* batch II liquid fraction fermented with birch sap; AS I (TKG)—*Avena sativa* batch I liquid fraction fermented with Tibetian kefir grains; AS II (BS)—*Avena sativa* batch II liquid fraction fermented with Tibetan kefir grains.

**Table 4 foods-11-02055-t004:** Inhibition zones of *A. sativa* extract after fermentation with Tibetan kefir grains and birch sap against Gram-positive and Gram-negative bacteria strains.

Bacterial Strains		Inhibition Zone ± SD, mm	
		*A. sativa* I (TKG)	*A. sativa* II (TKG)
Gram-positive	*S. aureus*	5.3 ± 0.10 b	6.5 ± 0.15 a
	*A. streptococci*	4.9 ± 0.60 c	5.8 ± 0.20 d
Gram-negative	*P. aeruginosa*	0.5± 0.10 e	0.5 ± 0.10 e
	*E. coli*	1.5 ± 0.60 f	1.2 ± 0.50 f

Values are expressed as mean and standard deviation calculated with triplicate determinations; different letters within the same column indicate significant differences (one-way ANOVA and Tukey’s HSD test, *p* < 0.05).

## Data Availability

Data is contained within the article and Appendix A.

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
