# Peer review of "The Biochemical Alteration of Enzymatically Hydrolysed and Spontaneously Fermented Oat Flour and Its Impact on Pathogenic Bacteria"

_foods, 2022, doi:10.3390/foods11142055_

Round 1

Reviewer 1 Report

The authors compared the biochemical changes of enzymatic hydrolysis and spontaneous fermentation of oat flour and its effect on pathogenic bacteria. The topic of this paper is suitable for the scope of the Journal. The results are interesting and the work is well performed. However, the paper should undergo extensive English editing. I have also some questions and comments listed below.
 1. The author lists too many experimental content in the abstract, and it is recommended to summarize the content appropriately and supplement the key data and results.
2. Line 160 Figure 2 caption is confusing.
3. Line 236  HPLC-SEC Is the abbreviation of "High-pressure liquid chromatography size exclusion"; "molecular mass sequencing" should be changed to "molecular mass distribution".

Author Response

The authors are grateful for the Reviewers report, which impactfully improves the manuscript. Below we answered your comments. Also, English editing and revision were performed, improving the manuscript's quality.

Point 1: The author lists too many experimental content in the abstract, and it is recommended to summarize the content appropriately and supplement the key data and results.

Answer 1: Thank you for your comment; we agree with it and modified the abstract, which included the results part (line 25-35)

Point 2: Line 160 Figure 2 caption is confusing.

Answer 2: Thank you for your comment; we modified the Figure 2 caption in the manuscript (line 164)

Point 3: Line 236  HPLC-SEC Is the abbreviation of "High-pressure liquid chromatography size exclusion"; "molecular mass sequencing" should be changed to "molecular mass distribution".

Answer 3: Thank you for your comment; authors’ agree and changed it in the manuscript (line 258)

Thank you,

Reviewer 2 Report

The manuscript (entitled Biochemical alteration of enzymatically hydrolyzed and spontaneous fermented oat flours and impact against bacteria’s ) has been written in systematic and comprehensive, but some parts need to be deepened for their discussion.

Comments:

Title: it should be clear for against pathogenic bacteria

Abstract:

A brief intro on antimicrobial from lactic acid bacteria fermentation should be there.

The objective is not clear, because antimicrobial was assessed for the fermented extract, which extract was obtained from enzymatic extraction.

The abstract of experimental design was not clear too.

Figure 1: Type of enzymes used should be appeared in the figure, in order to not misleading, and fermentation by LAB should be described in the figure.

Table 1: Enzymes used in the extraction (Batch I and II) should be stated as a note. The method of analysis, by NDF and ADF should be clear in the note for a self explaining table.

Line 131-132: The main non- 131 starch polysaccharide enzyme is cellulose (EC 3.2.1.4). Change cellulose to cellulase. Cellulose is its substrate.

The picture of SEM is not Figure 1, but Figure 2. The appearance of the SEM results are not discussed well. Figure 2 becomes Figure 3, and so on. The unit g or ml must be separated from the value, for example 100 g or 100 ml. For ml must be written as mL. This must be applied throughout the manuscript.

Some inconsistency in writing are found (for the unit and also for value, for example 70 000 kDa, and 7000 kDa).

Table 2, please explain EAE, GAE, TPC in the table note. The ratio for example 30:70 is not clear, is it between water and ethanol? Or ethanol and water?

Figure 3 (actually Figure 4), is not clear for (a) to (i) note.

The legend in Figure 3 should state the organic acids analyzed, are there lactic acid and acetic acid?

Method: the chemical standards for molecular weight measurement by HPLC-SEC are not clear. PCA statistical analysis is not stated in the method, but it is there in the result and discussion.

Write kDa, in Figure 4d.

Discussion on molecular weights and components inside the samples is still not sufficient.

Table 4 the inhibition zones, the positive control for the inhibition zone (using a commercial antibiotic) should be analyzed. The calculation , is it after reduced with 6 mm disk diameter?

The comprehensive discussion to gather or to link the phenolic and the antibacterial activity results is not there.

Author Response

The authors are grateful for the Reviewers report, which impactfully improves the manuscript. Please, see the attachment .

Also, English editing and revision were performed, improving the manuscript's quality.

Reviewer 3 Report

This study investigated biochemical alteration of enzymatically hydrolyzed and spontaneous fermented oat flours. The results indicated promising applications for developing functional products and components using bioprocessing as an innovative tool. I recommend minor revisions.

1.       At lines 130-131, the authors have concluded from change of the SEM morphologies by enzyme treatment that enzymes can access and disrupt the polymeric chains into smaller ones. Why does change of sizes on molecular level affect change of the SEM morphologies on micrometer scale?

2.       At lines 132, ‘cellulose’ should be revised to be ‘cellulase’.

3. English throughout the text should be checked and revised.

Author Response

Point 1 : At lines 130-131, the authors have concluded from change of the SEM morphologies by enzyme treatment that enzymes can access and disrupt the polymeric chains into smaller ones. Why does change of sizes on molecular level affect change of the SEM morphologies on micrometer scale?

Answer 1: Thank you for your comment. The breakdown of polymer chains takes place at the molecular level; then, all this is very clearly visible using SEM at micro/nano levels, where reduced particles' size can be seen by choosing a suitable microscale.

Pont 2: At lines 132, ‘cellulose’ should be revised to be ‘cellulase’.

Answer 1: Thank you, we correct it in the manuscript

Pont 3: English throughout the text should be checked and revised.

Answer 1: Thank you, we revised English throughout all manuscript.

Sincerely,